# SACoD: Sensor Algorithm Co-Design Towards Efficient CNN-powered Intelligent PhlatCam

## Abstract

There has been a booming demand for integrating Convolutional Neural Networks (CNNs) powered functionalities into Internet-of-Thing (IoT) devices to enable ubiquitous intelligent "IoT cameras". However, more extensive applications of such IoT systems are still limited by two challenges. First, some applications, especially medicine- and wearable-related ones, impose stringent requirements on the camera form factor. Second, powerful CNNs often require considerable storage and energy cost, whereas IoT devices often suffer from limited resources. PhlatCam, with its form factor potentially reduced by orders of magnitude, has emerged as a promising solution to the first aforementioned challenge, while the second one remains a bottleneck. Existing compression techniques, which can potentially tackle the second challenge, are far from realizing the full potential in storage and energy reduction, because they mostly focus on the CNN algorithm itself. To this end, this work proposes SACoD, a **S**ensor **A**lgorithm **Co**-**D**esign framework to develop more efficient CNN-powered PhlatCam. In particular, the mask coded in the PhlatCam sensor and the backend CNN model are jointly optimized in terms of both model parameters and architectures via differential neural architecture search. Extensive experiments including both simulation and physical measurement on manufactured masks show that the proposed SACoD framework achieves aggressive model compression and energy savings while maintaining or even boosting the task accuracy, when benchmarking over two state-of-the-art (SOTA) designs with six datasets on four different tasks. We also evaluate the performance of SACoD on the actual PhlatCam imaging system with visualizations and experiment results. All the codes will be released publicly upon acceptance.

## 1 Introduction

Recent CNN breakthroughs trigger a growing demand for intelligent IoT devices, such as wearables and biology devices (e.g., swallowed endoscopes). However, two major challenges are hampering more extensive applications of CNN-powered IoT devices. First, some applications, especially medicine- and biology-related ones, impose strict requirements on the form factor, especially the thickness, which are often too stringent for existing lens-based imaging systems. Second, powerful CNNs require considerable hardware costs, whereas IoT devices only have limited resources.

For the first challenge, lensless imaging systems (Asif et al., 2015; Shimano et al., 2018; Adams et al., 2017; Antipa et al., 2018; Boominathan et al., 2020) have emerged as a promising rescue. For example, PhlatCam (Boominathan et al., 2020) replaces the focal lenses with a set of phase masks, which encodes the incoming light instead of directly focusing it. The encoded information can be either computationally decoded to reconstruct the images or processed specifically for different applications. Such lensless imaging systems can be made much smaller and thinner, because the phase masks are smaller than the focal lens, and they can be placed much closer to the sensors and fabricated with much lower costs. For the second challenge, many recent works focus on designing CNNs with improved hardware efficiency, i.e., by applying generic neural architecture search (NAS) to find efficient CNNs.

As such, a naive way to address the two aforementioned challenges simultaneously is to introduce lensless cameras as the signal acquisition frontend and then apply NAS to optimize the backend CNN. However, such approaches would result in disjoint optimization that can be far from optimal. A generic NAS would treat the camera as given, and only optimize the CNN. Likewise, existing

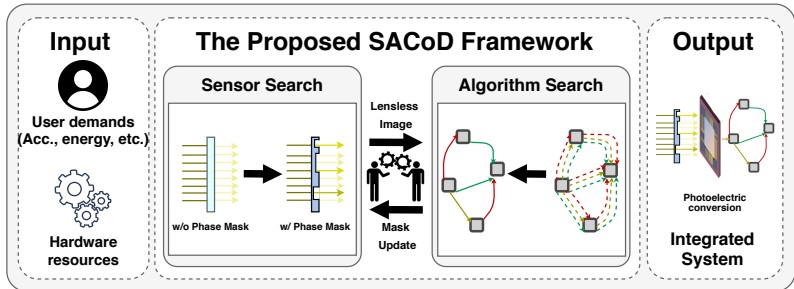

Figure 1: An overview of the proposed SACoD framework.

phase mask designs for lensless cameras treat the CNNs as given, and only optimize the masks. Such disjoint optimization fails to (1) take advantage of the masks' potential computational capacity, with which the NAS optimization can be fundamentally improved, and (2) perform end2end optimization.

It is shown in (Boominathan et al., 2020) that, under some assumptions, the phase masks in Phlat-Cam essentially perform 2D convolutions on the incoming lights, and the convolution kernel is encoded in the masks. Moreover, unlike other convolutional layers, the phase masks' convolutions are almost free – they do not consume additional energy, computation power, or storage, **regardless of** what value each mask takes. Therefore, we aim to incorporate the phase mask design into NAS to enable an end2end optimization of the sensing-processing pipeline, while exempting a portion of the pipeline from the efficiency penalties. Such co-designs are expected to achieve better tradeoffs between accuracy and efficiency.

To this end, we propose a Sensor Algorithm Co-Design (SACoD) framework to enable more energy-efficient CNN-powered IoT devices. While SACoD is general and can be applied to different sensing and intelligent processing systems, it is developed and evaluated in the context of PhlatCam (Boominathan et al., 2020) based imaging systems. Our main contributions are:

- We propose SACoD, a novel co-design framework that jointly optimizes the sensor and neural networks to enable more energy-efficient CNN-powered IoT devices. To our best knowledge, SACoD is the first to propose sensor algorithm co-design for CNN inferences.

- We develop an effective design of the optical layer to (1) exploit its potential computation capability and (2) enable co-search of the optical layer and backend algorithm. We then characterize the trade-off between accuracy and the required area of the corresponding imaging systems to demonstrate its effectiveness under practical size constraints.

- Extensive experiments and ablation studies validate that the proposed SACoD consistently achieves reduced hardware costs/area while offering a comparable or even better task accuracy, when evaluated over two SOTA lensless imaging systems on four tasks and six datasets. And part of the experiments are further evaluated with fabricated masks to validate SACoD's effectiveness in the physical measurement besides simulation.

## 2 RELATED WORKS

**Neural architecture search.** Recently NAS has attracted increasing attention. It eliminates the handcrafting process and automatically designs neural architectures. Existing NAS techniques can be divided into three categories, evolution-based, reinforcement-learning (RL)-based, and one-shot NAS. As the computational overheads of evolution- or RL-based approaches can be unacceptably high, many techniques (Brock et al., 2017; Cai et al., 2018a; Liu et al., 2017; 2018; Pham et al., 2018; Xie et al., 2018) have been proposed to reduce the searching cost, among which differentiable architecture search (DARTS) has gained intensive interests. While being conceptually general, SACoD in this paper adopts the DARTS method, where a super-network is optimized during search and the strongest sub-network is preserved and then retrained.

**Lensless imaging systems.** To eliminate the size or thickness burden caused by the lens, various lensless imaging systems have been developed. While lensless imaging systems have been widely used for capturing X-ray and gamma-ray (Dicke, 1968; Caroli et al., 1987), it is still in an exploring stage for visible spectrum uses (Asif et al., 2015; Shimano et al., 2018; Antipa et al., 2018; Boom-

inathan et al., 2020). In general, lensless imaging systems capture the scene either directly on the sensor or after being modulated by a mask element.

In this paper, we focus on a specific lensless imaging system based on phase masks called PhlatCam (Boominathan et al., 2020), which is a general-purpose framework to create phase masks that can achieve desired sharp point-spread-functions (PSFs). A phase mask modulates the phase of incident lights, and allows most of the light to pass through, providing a high signal-to-noise ratio. Hence, they are desirable for low light scenarios and photon-

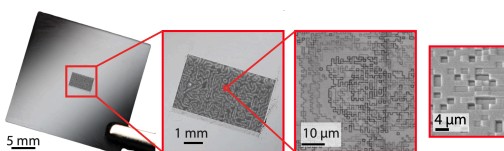

Figure 2: A fabricated phase mask used in the PhlatCam lensless imaging system (Boominathan et al., 2020).

limited imaging. Fig. 2 shows a fabricated phase mask, which is essentially a transparent material with different thicknesses at different locations. Based on this lensless imaging system, we develop our SACoD framework, aiming to enable more energy-efficient CNN-powered IoT devices.

**Sensor-algorithm co-training.** There have recently been some attempts that try to jointly optimize the sensor parameters and the neural network backend. For lens-based image systems, novel lens designs are introduced and trained concurrently with the neural network backend to jointly optimize for image reconstruction (Sitzmann et al., 2018), depth estimation (Chang & Wetzstein, 2019), and high-dynamic-range imaging (Metzler et al., 2019). Similar approaches have been applied to other imaging systems as well, including cameras with color multiplexing (Chakrabarti, 2016), Phase-Cam3D (Wu et al., 2019b), and Single Photon Avalanche Photodiodes cameras (Sun et al., 2020). Yet these methods still consider the neural network architecture as fixed, and do not explore the potential of applying the co-design principle to the neural architecture design.

## 3 THE PROPOSED SACoD FRAMEWORK

This section presents the SACoD framework. We will first outline the framework and explain why the optical layer can be considered as a convolutional layer, and then introduce how we implement SACoD's optimization algorithm. Finally, we describe our developed optical layer design.

**Framework setup.** The SACoD framework shown in Fig. 1 consists of two modules, an optical sensing frontend and a neural network backend. The coded mask of PhlatCam in the sensor and the backend are jointly optimized using a SOTA differential NAS algorithm (Liu et al., 2018), where the coded masks, together with the neural network weights, are regarded as network parameters.

Specifically, the first module, i.e. the optical sensing frontend, is denoted as $O(\cdot; \boldsymbol{m})$, where $\boldsymbol{m} = \boldsymbol{m}(x, y)$ denotes the phase mask values. The optical layer is based on the PhlatCam system (Boominathan et al., 2020). It receives the light signal from the object in front of the camera, processes the signal using the phase masks, and generates the sensor output. The second module, i.e. the neural network backend, is denoted as $F(\cdot; \boldsymbol{w}, \boldsymbol{\alpha})$, where $\boldsymbol{w}$ represents weights of the neural network, and $\boldsymbol{\alpha}$ parameterizes the architecture. The neural network backend receives the sensor signal and produces an output for the intended applications.

Formally, denote the light signal as $\boldsymbol{I}(x, y) \in \mathbb{R}^{H \times W \times 3}$, where $x$ and $y$ are coordinate indices and $H$ and $W$ represent the height and width of the range of light the camera can receive. The light signal contains RGB channel, and hence the last dimension is 3. Denote the signal received at sensor as $\boldsymbol{Z}(x, y) \in \mathbb{R}^{H' \times W' \times N}$, where $H'$ and $W'$ represent the height and width, and $N$ represent the number of channels. Denote $Y$ as the final output of the neural network backend. Then we have

$$\boldsymbol{Z} = O(\boldsymbol{I}; \boldsymbol{m}), \quad Y = F(\boldsymbol{Z}; \boldsymbol{w}, \boldsymbol{\alpha}). \tag{1}$$

The following subsections will introduce the form of $O(\cdot; \boldsymbol{m})$ and how to determine $\boldsymbol{m}$, $\boldsymbol{w}$, and $\boldsymbol{\alpha}$.

**The optical sensing frontend.** Assuming that the light signal $\boldsymbol{I}(x, y)$ comes from an object whose distance to the camera is $d$, and that the depth of the object is relatively small, it can be shown (Boominathan et al., 2020) that $O(\cdot; \boldsymbol{m})$ takes the following convolutional form:

$$\boldsymbol{Z}(x, y) = O(\boldsymbol{I}; \boldsymbol{m}) = \boldsymbol{p}(x, y; \boldsymbol{m}, d) * \boldsymbol{I}(x, y), \tag{2}$$

where $*$ denotes 2D convolution, $\boldsymbol{p}(x, y; \boldsymbol{m}, z)$ is called the *point spread function* of the phase mask, which is determined by the phase mask $\boldsymbol{m}(x, y)$ and the distance $d$. Once we optimize the PSF, the

phase masks are designed for the PSF and a chosen $d$. The fabricated mask then produces the PSF at the given $d$. For the fabricated system shown in 4.5, $d$ is set to be 2 mm for making our system much thinner than conventional cameras (thickness ranges between 7-20 mm). The mask is fixed at distance $d$ to the sensor during operation, and thus the convolution property will continue to hold. According to Eq. equation 2, the optical layer can be regarded as a special convolutional layer. Note that one phase mask can only perform a single-channel convolution with a *positive kernel*. It takes two phase masks to implement a single-channel convolution with a real-valued kernel, where one implements the positive part of the kernel and the other implements the negative part. Therefore, in order to construct a convolutional layer with three channels and real-valued kernels, we need six masks in the imaging system. Also, the input light has three color channels (R, G, and B), and each phase mask operates on all the color channels. Therefore, a three-channel convolution will produce a total of nine feature maps (FMs).

We propose three different designs for using the rendered FMs, as shown in Fig. 3. Specifically, the design in Fig. 3 (a) accumulates the FMs across the same color and outputs a 3-channel FM, which is still in an RGB shape; the design in Fig. 3 (b) accumulates the FMs from one mask across three colors and outputs a 2-channel FM; and the design in Fig. 3 (c) simply concatenates all the FMs from different colors. These three different designs extract different amounts or types of information from the scene which are then passed to the following neural network.

Experiments under various settings show that SACoD based on Fig. 3 (a) achieves higher accuracies than those based on Fig. 3 (b), and similar accuracies as the designs based on Fig. 3 (c). We conjecture the reason is that design (a) applies independent transformations on the RGB channels to maintain the original channel-wise discriminative information; and maintaining all the information as design (c) does not contribute to higher accuracy than design (a), since the accumulated information across the same color has provided sufficient information for the following processing under the constrained number of masks. So in our experiments, we adopt the optical layer design (a).

**The SACoD formulation and algorithm.** This subsection introduces the formulation and optimization of SACoD which aims to simultaneously optimize the phase mask $m$, and the neural network's architecture $\alpha$, and the neural network's weights $w$. Formally, SACoD aims to solve:

$$\min_{\alpha} \mathcal{L}_{val}\left(m^*(\alpha), w^*(\alpha), \alpha\right) + \lambda\mathcal{L}_e(\alpha), \tag{3}$$

$$m^*(\alpha), w^*(\alpha) = \arg\min_{\{m,w\}} \mathcal{L}_{tr}(m, w, \alpha). \tag{4}$$

$\mathcal{L}_{tr}$ and $\mathcal{L}_{val}$ are task-specific performance losses evaluated on the training and validation set, respectively, $\mathcal{L}_e$ is the efficiency loss, *e.g.* model size, computational cost, or energy consumption, and $\lambda$ is the tuning parameter trading-off the accuracy and efficiency. Following the same parameterization scheme in DARTS (Liu et al., 2018), $\alpha$ denotes the weights of different candidate operations.

There are two major modifications compared to the original DARTS framework. The first difference is that the efficiency loss $\mathcal{L}_e$, measured by the sum of each layer's FLOPs weighted by the network parameter $\alpha$, is introduced. More importantly, the second and major difference is that the phase mask $m$ is optimized jointly in the framework. It is worth pointing out that although mathematically similar, $m^*$ and $w^*$ have different degrees of dependencies on $\alpha$. $w^*$ is directly impacted by $\alpha$ because $\alpha$ governs which subset of the $w$ is ultimately used. $m^*$ is only indirectly influenced by $\alpha$. Therefore, incorporating $m$ will significantly improve the tradeoff between performance and model complexity. Besides, SACoD is naturally compatible with other NAS methods. We adopt differential NAS for the fast generation of the optical mask and network. When using other NAS methods, e.g., reinforcement learning-based NAS (Zoph & Le, 2017), we still observe similar system performance (within 0.3% accuracy on CIFAR-100), but the search time increases to 8 GPU-days from 0.5 GPU-days.

The whole co-design process can be divided into two stages, a searching stage and a training stage. In the searching stage, we apply the alternate gradient descent of Eqs. equation 3 and equation 4 to search for the optimal network architecture $\alpha^*$. In the training stage, the optimal mask and weights are determined by optimizing Eq. equation 4 conditioning on the optimal network architecture $\alpha^*$.

## 4 EXPERIMENTS

This section presents evaluation results of SACoD applied to PhlatCam lensless imaging systems. We first describe the experiment settings, and then benchmark SACoD over SOTA lensless

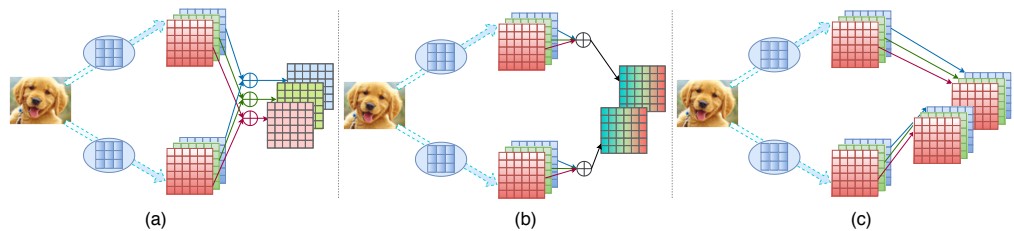

Figure 3: An illustration of the proposed three different optical layer designs.

imaging systems on standard classification tasks, IoT applications, and other vision tasks in Sections 4.2, 4.3, 4.4, respectively. Finally, we provide ablation studies of SACoD in Section 4.6.

## 4.1 EXPERIMENT SETUP

**Optical layer constraints.** As mentioned, the optical layer first performs convolutional operations on the input scene optically, the outputs of which are then processed by the following neural network. The physical device construction imposes design constraints on the optical layer design. Specifically, since the phase mask is placed closer to the sensor, the optically achievable kernel size cannot be arbitrarily small. Here, we adopt kernel sizes that are not smaller than 7x7. Additionally, since all the designed masks are sharing the same sensor area, the number of designed masks cannot be large due to the limited sensor area. Here, we constrain the number of masks to be no more than 6.

**Algorithm setting.** Datasets: we evaluate SACoD on a total of four vision tasks with six datasets: two classification datasets CIFAR-10/100, two IoT datasets including FlatCam Face (Tan et al., 2019) and Head Pose Detection (Gourier & Crowley, 2004), one segmentation dataset Cityscapes (Cordts et al., 2016), and one unsupervised image translation dataset horse2zebra (Zhu et al., 2017). The same and standard data augmentation methods (e.g., random crop and normalization) are adopted for both SACoD and the baselines.

Baselines: we benchmark SACoD over two SOTA lensless imaging systems:

- Gabor-mask System: we fix the optical layer to be the Gabor-mask (Chen et al., 2016) and search for networks using the same NAS method as SACoD.
- Co-train System: we fix the network to be a SOTA IoT CNN MobileNetV2 (Sandler et al., 2018) and jointly train the optical layer and the backend network.

Efficiency metrics: we consider both FLOPs (Floating Point Operations) and energy cost based on **real-device** measurements as the efficiency metrics. Specifically, we adopt an NVIDIA JETSON TX2 (NVIDIA Inc.), a popular IoT GPU, as the target platform, which is connected to a laptop with the real-time energy cost being obtained via the sysfs (Patrick Mochel and Mike Murphy.) of the embedded INA3221 (Texas Instruments Inc.) power rails monitor.

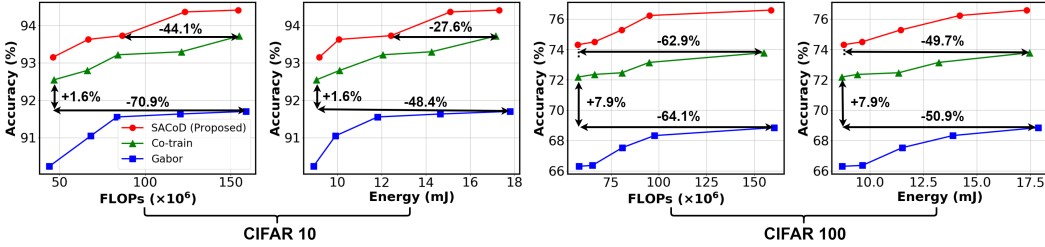

Figure 4: Accuracy vs. FLOPs/energy trade-offs of SACoD and the baselines on CIFAR-10/100.

## 4.2 SACoD OVER SOTA IMAGING SYSTEMS ON CLASSIFICATION TASKS

**Search and training setting.** To search neural networks on CIFAR-10/100 for both the SACoD and Gabor-mask systems, we quantize all the operations to 8-bit using a SOTA quantization training method (Banner et al., 2018), which is a common practice considering the constrained sources on IoT devices. We adopt the search settings in (Wu et al., 2019a) with minor changes discussed in appendix. Specifically, we search for 50 epochs with 64 batch size, and update the supernet weights

on half of the training dataset using an SGD optimizer with a 0.9 momentum, and an initial learning rate of 0.025 with the cosine decay, and update network architecture parameters on the other half of the training dataset using an Adam optimizer with 0.9 momentum, and a fixed learning rate of 3e-4. For training the derived network architectures from scratch, we adopt an SGD optimizer with 0.9 momentum, and an initial learning rate of 0.01 with cosine decay for 600 epochs with 96 batch size.

To benchmark SACoD over SOTA imaging systems, we fix the number of masks to be six among all the settings, and then study their accuracy under different FLOPs and energy costs. We control the FLOPs of the SACoD and Gabor-mask systems by controlling $\lambda$ in Eq. equation 3 and that of the Co-train system by changing the width multiplier (Howard et al., 2017).

**Results analysis.** Fig. 4 shows the trade-off between the accuracy and required hardware costs in terms of both FLOPs and energy cost for the SACoD and the two baseline lensless imaging systems on CIFAR-10/100. We can observe that SACoD consistently requires reduced FLOPs or energy cost while achieving a comparable or higher accuracy over the baselines. On CIFAR-10, SACoD achieves 44.1% and 70.9% reduction in FLOPs, and 27.6% and 48.4% reduction in energy, while offering a +0.01% and +1.58% higher accuracy, compared with the Co-train and Gabor-mask baselines, respectively; On CIFAR-100, SACoD reduces the FLOPs by 62.9% and 64.1%, and energy cost by 49.7% and 50.9%, while achieving a +0.71% and +7.92% higher accuracy, compared to the Co-train and Gabor-mask baselines, respectively. This set of experiments validate that the end-to-end optimization engine in SACoD indeed can lead to superior performance in both task performance and hardware efficiency.

Considering that the form factor or area is another influential design factor in lensless IoT imaging systems, we evaluate SACoD over the baselines in terms of the trade-off between accuracy and area by controlling the number of masks in the optical layer, and summarize the results in Fig. 5. We can see that the proposed SACoD achieves the best accuracy-area tradeoffs among all the designs under the same number of masks (and thus area) and the same model size. In particular, SACoD achieves 60.0% and 80.0% reduction in area while offering a 0.01% and 1.05% higher accuracy, compared with the Co-train and Gabor-mask baselines, respectively.

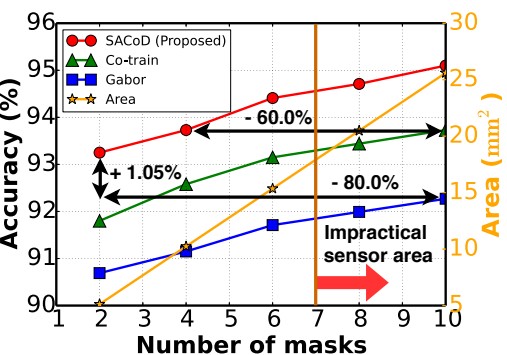

Figure 5: No. of masks vs. accuracy and sensor/mask area on CIFAR-10.

### 4.3 SACoD over SOTA imaging systems on IoT applications

Here we benchmark SACoD over the SOTA baselines on two IoT applications (including FlatCam Face recognition (Tan et al., 2019) and Head-pose task (Gourier & Crowley, 2004)) to evaluate its effectiveness on real-world IoT tasks. In this set of experiments, we further constrain FLOPs of the derived neural networks to see if SACoD is still applicable to extremely energy-constrained scenarios. As shown in Fig. 6, we can see that again SACoD consistently outperforms the baselines under all settings in terms of accuracy-cost tradeoffs. Specifically, compared with the Co-train baseline, SACoD achieves 59.5% and 57.1% reduction in FLOPs, 32.9% and 30.1% reduction in energy cost with a +0.11% and +0.07% higher accuracy, on the Flatcam Face and Head-pose tasks, respectively. Meanwhile, compared with the Gabor-mask baseline, SACoD shows better scalability to more energy-constrained scenarios. In particular, when the FLOPs or energy constraint is extremely low, SACoD achieves an 8.75% and 5.85% higher accuracy, under the same FLOPs/energy cost on the Flatcam Face and Head-pose tasks. These results show the effectiveness of SACoD extends to read-world IoT applications and the superior scalability of SACoD over SOTA IoT imaging systems.

### 4.4 SACoD over SOTA imaging systems on other vision tasks

Considering the diverse applications of IoT devices, we also evaluate SACoD on other vision tasks including one segmentation task and two unpaired image-to-image translation tasks, which require a more challenging tradeoff on CNN-powered intelligent IoT devices. We show the quantitative comparison, visualization, and detailed experiment settings in the appendix. Note that segmentation tasks are commonly evaluated in terms of **both** FID (Heusel et al., 2017) and image visualization as

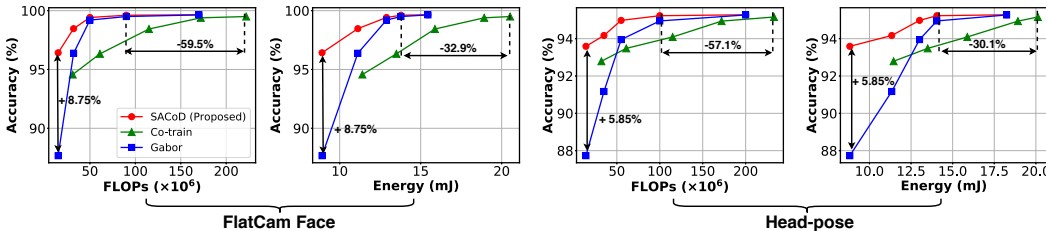

Figure 6: Accuracy vs. FLOPs/energy of SACoD over the baselines on two IoT tasks.

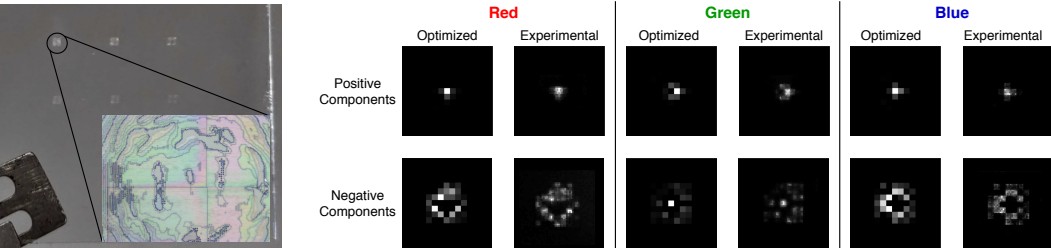

Figure 7: Physical Masks     Figure 8: PSF comparison between optimized and experimental ones

Table 1: Performance of SACoD on the actual PhlatCam imaging system.

| Datasets | Setting | Accuracy (%) | Gap (%) |
|----------|---------|--------------|---------|
| CIFAR10 | Optimized | 94.43 | 4.41 |
| | **Experimental** | **90.02** | |

the former cannot always capture the image quality. From the results, SACoD requires the smallest FLOPs under all the six cases, and it shows that SACoD provides notably the best visualization effect among all the methods. In particular, compared to the Gabor-mask baseline, SACoD achieves $12.0\% \sim 50.1\%$ reduction in FLOPs and $+1.26 \sim +20.27$ improvement in FID, while providing notably better visualization effect; compared to the Co-train baseline, SACoD reduces the FLOPs by $92.56\% \sim 93.4\%$ and offers notably better visualization effect, while achieving better FID on the zebra2horse dataset. Note that although the Co-train baseline achieves better FID than SACoD on the horse2zebra dataset, its visualization images (see Fig. 12 the appendix) suffer greatly from color shifts, distortion and chase-board effect, similar to those of the Gabor-mask baseline.

## 4.5 EVALUATION ON PHYSICAL FABRICATED MASKS

To evaluate the performance of SACoD on the actual PhlatCam imaging system besides simulation, we fabricate the actual masks with the PSF from the optimized optical layer. We first visualize the fabricated masks and compare them with the optimized PSF. Fig. 7 shows 6 fabricated masks setting on CIFAR-100 with a microscope image, in which the top row is positive masks and bottom ones are negative, and columns from left to right represent RGB-channels respectively. Then we compare the experimental PSF generated by the fabricated masks with the optimized PSF in Fig. 8. From the comparison, we find the experimental PSF basically keeps the original shape, although the brightness of some pixels is changed. We further compare the achieved accuracy based one simulated and fabricated masks on CIFAR-10. As shown in Table 1, the accuracy gap is 4.41% which is resulting from possible alignment issues and other experimental error sources in the mask fabrication process and has been shown by other optical computing systems, i.e., (Lin et al., 2018).

## 4.6 ABLATION STUDIES ON SACoD

**Influence of mask flexibility.** To better understand SACoD's superior performance, we perform ablation experiments where the optical mask is fixed at either or both of the search and training stages. The experiment setting is the same as described in Section 4.2, in which the models are all under comparable model sizes (1.0 M), and the pre-trained optical mask is obtained by co-training with MobileNetV2 (Sandler et al., 2018). We explore four cases: (mask) fixed during both the search and training stages, fixed only during the training stage, fixed only during the search stage, and unfixed

Table 2: Ablation: whether to fix the optical mask during search and training using small models.

| | CIFAR-100 | | | | | CIFAR-10 | | | | |
|---|---|---|---|---|---|---|---|---|---|---|
| Search | fixed | unfixed | fixed | **unfixed** | Improv. | fixed | unfixed | fixed | **unfixed** | Improv. |
| Training | fixed | fixed | unfixed | **unfixed** | | fixed | fixed | unfixed | **unfixed** | |
| Acc (%) | 67.45 | 67.69 | 68.03 | **69.64** | +2.19 | 92.02 | 92.37 | 92.84 | **92.95** | +0.93 |

during both (i.e., SACoD). From the results in Table 2 we can see that (1) fixing the mask during both the search and training stages causes the biggest accuracy drop (more than 0.93% on CIFAR-10 and CIFAR-100 datasets); and (2) fixing the mask at only one of the stages would also significantly impact the performance. This study verifies that the success of the co-design principle can be ascribed to two causes. First, with the mask included as hyperparameters, the optimization (during fine-tuning) can reach a better minimum. Second and more importantly, the trainable mask can assist finding a *better architecture* because the neural network backend can entrust more computation responsibilities to the 'costless' optical layer and thus can be made more efficient.

**Effectiveness of the optical layer.** To further explore the reason behind the success of SACoD, we compare the discriminative power of the features captured by the optical layers of SACoD and the Gabor-mask baseline. Specifically, following (Suau et al., 2020), we average the optical layer's activations over the output channels to obtain a vector and use the corresponding softmax value as the feature distribution for each input image. We then calculate the KL divergence between the feature distribution from different classes to see how discriminative the features are. Fig. 9 visualizes the average KL divergence (over 100 random selected im-

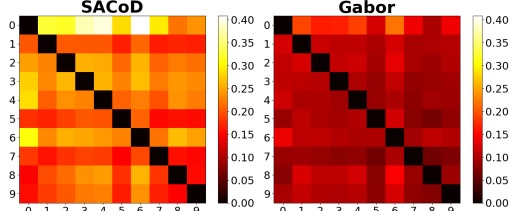

Figure 9: KL divergence of the output distribution between different classes captured by the searched optical layer of SACoD and Gabor-mask on CIFAR-10, where the x-axis and y-axis are the class id, and the heatmap value denotes the magnitude of KL divergence.

ages) between every two classes on the test dataset of CIFAR-10. We can see that the feature distribution difference of SACoD between different classes is notably and consistently larger than that of the Gabor-mask baseline, which further verifies that the optical layer of SACoD undertakes more computations to extract the discriminative information from the input, so as to save the computations for the neural network backend.

Considering the potential defection when fabricating the PhlatCam sensors, we study the robustness of the SACoD, Co-train, and Gabor-mask systems when injecting noises into the masks. To simulate the noise effect, we sample noises from a standard normal distribution and add it to the masks; after that, we finetune the backend network for restoring the accuracy until convergence. By controlling the standard deviation ($\sigma$) of the standard normal distribution, i.e., the noises' dynamic range, we evaluate the robustness of the three lensless systems under different noise intensity and summarize the results. We can observe that (1) as the $\sigma$ increases (i.e., noises' intensity increases), the restored accuracy of all the three lensless systems decreases while the accuracy of SACoD is consistently higher than that of both the Co-train and Gabor-mask baselines; and (2) the Co-train baseline is the most sensitive to the noises injected to the masks, and suffers from up to 2.71% accuracy loss, the Gabor-mask baseline shows the least accuracy degradation (1.19%) under noises yet offering the lowest overall accuracy, and our proposed SACoD achieves a favorable tradeoff with the highest task accuracy and a medium level accuracy drops under noises. More details are provided in appendix.

## 5 CONCLUSION

We propose SACoD, a sensor algorithm co-design framework, to enable more energy-efficient CNN-powered IoT devices based on PhlatCam. A novel end-to-end co-search algorithm is presented to jointly optimize the coded mask of PhlatCam in the sensor and the neural network backend. Extensive experiments and ablation studies validate the superiority of the proposed SACoD in terms of both task performance and hardware efficiency as well as the optical layer effectiveness, when evaluated over SOTA lensless imaging systems on various tasks and datasets.

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

## A    EXPERIMENT DETAILS FOR SEARCHING ON CIFAR-10/100

For the experiments on CIFAR-10/10, we adopt the same search space as FBNet (Wu et al., 2019a) except each group's stride settings, i.e., a set of blocks with the same number pf output channels, for adapting to the resolution of images on CIFAR-10/100. In particular, we follow the strides settings for MobileNetV2 on CIFAR-10/100 as described in (Wang et al., 2019), which is $[1, 1, 2, 2, 1, 2, 1]$ for all the seven groups. In addition, following (Wu et al., 2019a), we apply a gumbel softmax on each architecture parameter option's contribution weights to the supernet, where the initial temperature is 3 and decays by 0.92 at the end of each epoch.

## B    MORE ABLATION STUDIES FOR SACoD: ROBUSTNESS AGAINST NOISE

Considering the potential defection when fabricating the PhlatCam sensors, we study the robustness of the SACoD, Co-train, and Gabor-mask systems when injecting noises into the masks. To simulate the noise effect, we sample noises from a standard normal distribution and add it to the masks; after that, we finetune the backend network for restoring the accuracy until convergence. By controlling the standard deviation ($\sigma$) of the standard normal distribution, i.e., the noises' dynamic range, we evaluate the robustness of the three lensless systems under different noise intensity and summarize the results in Fig. 10. We can observe that (1) as the $\sigma$ increases (i.e., noises' intensity increases), the restored accuracy of all the three lensless systems decreases while the accuracy of SACoD is consistently higher than that of both the Co-train and Gabor-mask baselines; and (2) the Co-

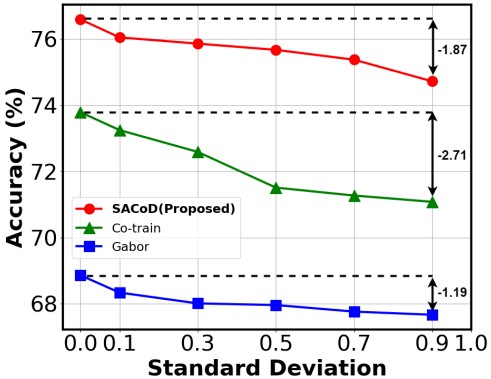

Figure 10: The recovered accuracy after finetuning the SACoD, Co-train, and Gabor-mask systems subject to different noise levels in the masks.

train baseline is the most sensitive to the noises injected to the masks, and suffers from up to 2.71% accuracy loss, the Gabor-mask baseline shows the least accuracy degradation (1.19%) under noises yet offering the lowest overall accuracy, and our proposed SACoD achieves a favorable tradeoff with the highest task accuracy and a medium level accuracy drops under noises.

## C    EXPERIMENT RESULTS WHEN USING DIFFERENT MASK DESIGNS

As illustrated in Fig. 3 of the main content, three different mask designs can be adopted. We here provide more experiment results when using the three mask designs for SACoD. Specifically, we co-train MobileNetV2 with the three different mask designs on CIFAR-10/100 and find that the design with the masks in Fig. 3 (a) (in the main content) achieves a 0.86% and 1.78% higher accuracy compared with that based on the masks in Fig. 3 (b), while offering a comparable accuracy (0.01% and -0.02%) with a 66.67% reduced rendered feature maps, on CIFAR-10 and CIFAR-100, respectively. This set of experiments motivates us to adopt the masks' design based on the one in Fig. 3(a) for experiments shown in the main content.

## D    SACoD VS. LENS-BASED SYSTEMS

Here we provide experiments for benchmarking SACoD over lens-based systems under the same search space (Wu et al., 2019a) and datasets (CIFAR-10/CIFAR-100). For designing the lens-based systems, we release the FLOPs constraints for the first layer of the network, i.e., removing the optical layer and its associated constraints, and search for the optimal network given the search space and datasets. We find that under a slightly reduced FLOPs (154M FLOPs vs. 158M FLOPs), SACoD achieves a 0.39% and 0.62% lower accuracy on CIFAR-10 and CIFAR-100, respectively, while reducing the thickness of the imaging systems by 10× which makes it possible to be integrated into more IoT applications. This set of experiments show that our proposed SACoD can offer similar

task performance and hardware efficiency as compared to lens-based systems, while being able to shrink the thickness of the system by one order.

# E   More details and results on the unpaired image-to-image translation tasks

In this subsection, we provide more details for the experiments on the unpaired image-to-image translation tasks, results of which are summarized in Table 2 and Fig. 7 of the main content.

## E.1   The search space and method

For the search space, we drew inspiration from existing works and build a sequential supernet with hardware-friendly regular connection patterns where each candidate operators are sequentially connected. For the search method, we adopt a SOTA differential neural architecture search (NAS) method (Liu et al., 2018) to search for both the operators and their widths to maximize efficiency improvements.

**Search for the operators.** The supernet consists of a total of nine operators from the following options:

- The Conv layer: Conv 1×1,    Conv 3×3;
- The residual block (i.e., 2 Conv 3×3 layers with a skip connection);
- The depthwise (DW) block (Conv 1 ×1 + DW Conv 3×3 + Conv 1×1 with a skip connection).

The above options cover popular and efficient building blocks in SOTA GAN generators (Zhu et al., 2017; Shu et al., 2019; Gong et al., 2019). In particular, following SOTA differential NAS search method, we use the architecture parameter $\alpha_{ij}$ to control the probability of choosing the $j$-th operator in the $i$-th layer, and treat its softmax values as its contributing weight to the supernet.

**Search for the widths.** Considering our target IoT applications, we also search for the widths (i.e., the number of output channels) of each operator for more efficient GAN models. Naively building a set of independent convolutional kernels with different widths for each operator is not practical due to the required large memory consumption. We thus build a *superkernel* with a maximal width, and then search for the expansion ratio $\phi$ to make use of only a subset of the *superkernel*'s input/output dimensions. In particular, we set $\phi \in \left[\frac{1}{3}, \frac{1}{2}, \frac{3}{4}, \frac{5}{6}, 1\right]$ and use the architecture parameter $\gamma_i$ to control the probability of choosing each expansion ratio for the $i$-th layer; Meanwhile, we apply gumbel softmax to approximate differentiable sampling for $\phi$ based on $\gamma$ as (Cai et al., 2018b). Therefore, during the searching process only one expansion ratio will be activated in one iteration, saving both the required memory and computational costs.

## E.2   Proxy Task

We search for an efficient generator via distillation as formulated below:

$$\min_{G, \alpha, \gamma} \frac{1}{N} \sum_{i=1}^{N} d(G(x_i, \alpha, \gamma), G_0(x_i)) + \lambda F(\alpha, \gamma). \tag{5}$$

Where $d(\cdot, \cdot)$ is a distance metric for the knowledge distillation (Hinton et al., 2015) between $G$ (the searched generator) and $G_0$ (the original generator in CycleGAN (Zhu et al., 2017)), which is the perceptual loss (Johnson et al., 2016) in our case, $F$ is the computational budget determined by the network architecture (FLOPs in our case), and $\alpha$ and $\gamma$ are the architecture parameters controlling the operator and width of each layer, respectively. Note that, the objective function in Eq. 5 is independent of any trained discriminators since in practice the discriminator is often discarded after the generator is trained and thus not necessarily available when compressing the generator.

## E.3   More visualization results

Here we show more visualization results of SACoD as compared with the two lensless baselines on the unpaired image-to-image translation tasks (horse2zebra and zebra2horse) in Fig. 12. We can

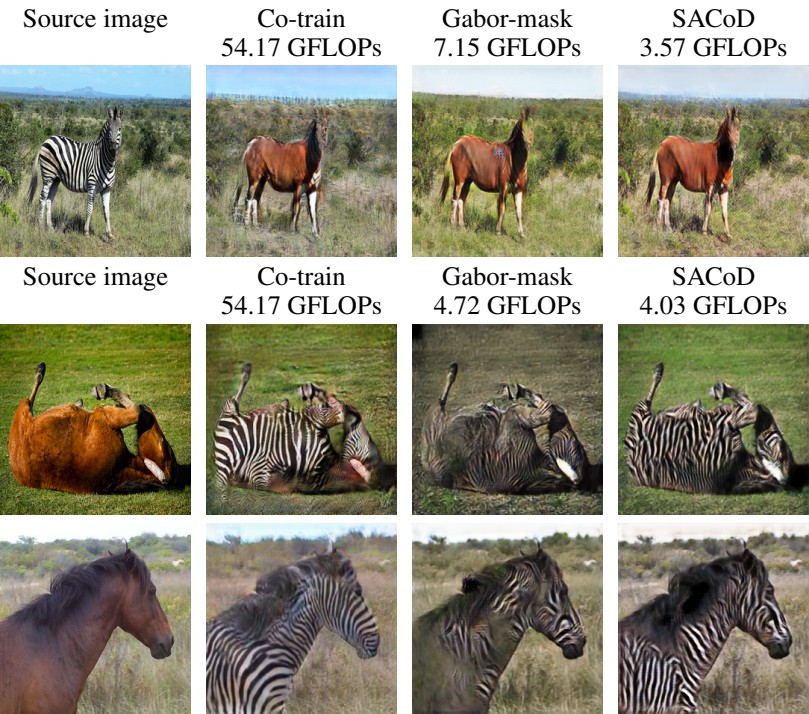

Figure 11: More visualization examples on the zebra2horse (row 1) and horse2zebra (row 2~3) tasks when using six masks. Columns from left to right: source images, and translation results of the Co-train, Gabor-mask, and SACoD methods, respectively, where the required FLOPs of each method on each task are annotated above the images.

Table 3: SACoD over SOTA baselines on a segmentation task with the Cityscapes dataset.

| Setting | Methods | 2 masks | | 4 masks | | 6 masks | |
|---|---|---|---|---|---|---|---|
| | | mIOU (%) | GFLOPs | mIOU (%) | GFLOPs | mIOU (%) | GFLOPs |
| Segmentation -Cityscapes | Co-train | 69.0 | 435.0 | 69.6 | 435.0 | 68.8 | 435.0 |
| | Gabor-mask | 65.8 | 45.64 | 66.1 | 38.32 | 67.3 | 36.34 |
| | **SACoD** | **69.8** | **36.17** | **70.4** | **33.56** | **71.6** | **29.51** |
| | **SACoD Improv.** | **+0.80~+4.0** | **20.7%~91.7%** | **+0.80~+4.3** | **12.4%~92.3%** | **+2.8~+4.3** | **18.8%~93.2%** |

see that again the Gabor-mask baseline suffers greatly from color shift and distortion especially on the horse2zebra task, and the proposed SACoD provides better or competitive visualization effects with high-contrast textures compared with the Co-train baseline while achieving a 92.6% reduction in FLOPs.

## E.4 SACoD OVER SOTA IMAGING SYSTEMS ON OTHER VISION TASKS

**Segmentation tasks.** Real-time segmentation on resource-constrained platforms have growing demands in many applications such as medical image segmentation, so we explore its combination with PhlatCam lensless imaging systems for scenarios with constrained imaging environments. To search for efficient segmentation networks, we adopt the search space and search method of a SOTA NAS method (Chen et al., 2019). For a fair comparison, all the searching and training settings follow the original ones in (Chen et al., 2019). Specifically, the co-train baseline adopts the DeepLabV3 (Chen et al., 2017) model with a ResNet50 (He et al., 2016) backbone. Table 3 with 2048x1024 images show that under all the mask constraints SACoD achieves the highest mean Intersection Over Union (mIOU) while requiring the smallest FLOPs. Specifically, SACoD achieves a 0.8%~4.3% higher mIOU and 12.4%~93.2% reduction in FLOPs over the Co-train and Gabor-mask baselines, respectively, under all the mask settings.

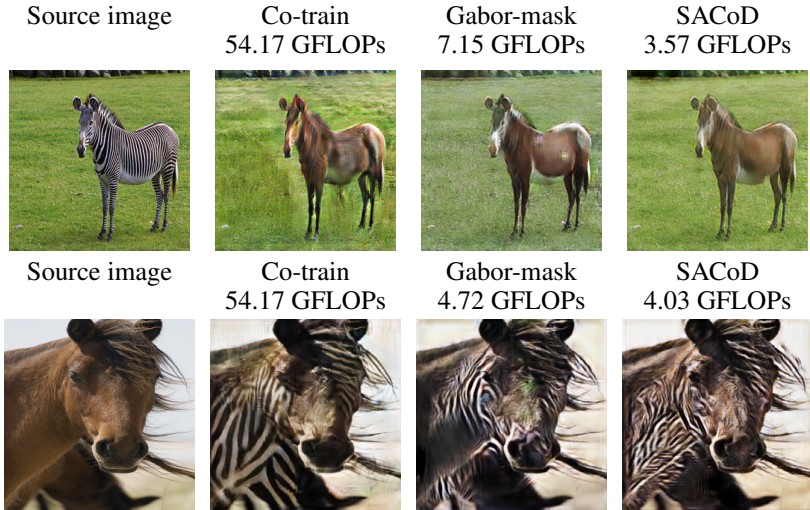

| Source image | Co-train 54.17 GFLOPs | Gabor-mask 7.15 GFLOPs | SACoD 3.57 GFLOPs |

| Source image | Co-train 54.17 GFLOPs | Gabor-mask 4.72 GFLOPs | SACoD 4.03 GFLOPs |

Figure 12: Visualizations on the zebra2horse (rows 1) and horse2zebra (row 2) tasks with six masks. Columns from left to right: source image, translation results for the Co-train, Gabor-mask and SACoD methods, respectively. FLOPs of each method on each task are annotated above the images.

Table 4: SACoD over SOTA baselines on unpaired image-to-image translation tasks.

| Setting | Methods | 2 masks | | 4 masks | | 6 masks | |
|---|---|---|---|---|---|---|---|
| | | FID | GFLOPs | FID | GFLOPs | FID | GFLOPs |
| zebra2horse | Co-train | 147.03 | 54.17 | 140.70 | 54.17 | 139.83 | 54.17 |
| | Gabor-mask | 137.79 | 6.89 | 141.11 | 5.04 | 145.87 | 7.15 |
| | **SACoD** | **136.35** | **5.93** | **136.41** | **3.89** | **138.23** | **3.57** |
| horse2zebra | Co-train | 66.82 | 54.17 | 61.21 | 54.17 | 68.26 | 54.17 |
| | Gabor-mask | 91.87 | 5.87 | 106.27 | 4.34 | 88.36 | 4.72 |
| | **SACoD** | **89.80** | **3.70** | **86.00** | **3.82** | **87.10** | **4.03** |

**Unpaired image-to-image translation tasks.** To apply GAN (Goodfellow et al., 2014) to lensless imaging systems under constrained imaging environments, we evaluate SACoD on unpaired image-to-image translation tasks (Zhu et al., 2017), one of the most popular applications of GAN.

Experiment settings: since no prior NAS works search for image-to-image GANs, we design a new search space (see details in the supplement) targeting extremely efficient GAN architectures. Following the settings of CycleGAN (Zhu et al., 2017), we keep the stem and header structures in the generator and search for both the operator type and channel width within the supernet which include nine sequential operators. In addition, we use the knowledge distillation (Hinton et al., 2015) between the searched architecture and the pre-trained CycleGAN generator as the loss function. More search settings are in the supplement. The Co-train baseline system consists of an original CycleGAN following the optical layer.

