# OpenReview forum: "SACoD: Sensor Algorithm Co-Design Towards Efficient CNN-powered Intelligent PhlatCam"
_ICLR.cc/2021/Conference — Reject_

### Official Review · AnonReviewer1 · 2020-10-25
**Review on Paper2602**

**Rating:** 6
**Confidence:** 3

**Review:**

Summary

This paper presents a method called SACoD to develop a more efficient CNN-powered Phlatcam. The proposed method optimizes both the PhlatCam sensor and the backend CNN model simultaneously.  That is, the coded mask in Phlatcam and neural network weights are regarded as learnable parameters. The coded mask (the optical layer) can be considered as a special convolution layer. As a result, it achieves energy saving, model compressing as well as good accuracy. Extensive experiments and ablation studies are presented to show the effectiveness of the method.

Overall,  I think the paper is interesting and properly designed. I also believe that various experiments and analyses of this paper can influence future related studies. However, I hope many experiments using raw images obtained from the real-world lensless imaging system will be included.

Strength

This paper is the first attempt to optimize both sensor and CNN-model at the same time. The derivation and formulation look interesting.


Comments and Weakness

In figure3, the input seems like a normal RGB image. Also, data in datasets such as CIFAR10 and Cityscape are common RGB images. Then, wasn't it applied to the real image obtained using the lensless imaging system? Even in Table 1, reported results were from common RGB images in CIFAR-10. In both optimized and experimental results, source images for testing are not real-world data from lensless imaging systems.

Is there any comparative test of the three methods in Figure3? Is there a reason why they are mentioned? The reasons why design (a) was chosen are mentioned in the text, it is also better to show them experimentally.

To what level can the Flops and energy be lowered based on the same baseline CNN model, and is it enough to put it on the real-world IoT system?

It would be better to compare the proposed method with more than two baseline methods. For example, I wonder what happens when regular CNN without optical layers is used. Also, it would be good to have an experiment on various types of mask filters except for Gabor.

About table2, What is the reason for the big difference in CIFAR-100 while there is not much increase in CIFAR-10?

Font sizes in Figure4 and Figure 6 are too small to read.

---

> ### Author Response · Authors · 2020-11-25
> **Response to Reviewer 1: Part 1**
>
> Thank you very much for recognizing our work with “various experiments and analyses of this paper can influence future related studies” and for your valuable suggestions. Below are our answers to you questions:
>
> (Here's the Part 1 of our responses, see Part 2 for more responses.)
>
> **1. Experiments using raw images obtained from the real-world lensless imaging system.**
>
> We will follow your suggestions and add more experiments based on raw images collected from fabricated masks in the final version.
>
> First, we want to clarify that the experiments in Table 1 of our manuscript is based on raw images obtained from the real-world prototyped lensless imaging system. We capture the real measurements of the dataset (e.g. CIFAR-10) by displaying dataset images on a monitor display and capturing using our experimental prototype. The CMOS sensor in our prototype has a Bayer RGB filter array on the pixels and so the sensor measurements after the mask can be split to different raw RGB color channels. Hence, our raw measurements have RGB channels as shown in Figure 3 of our manuscript.
>
> Second, we further fabricated the Gabor mask and tested its achieved accuracy following the same setting as that for Table 1 of our manuscript. We can observe that SACoD generated designs lead to lower accuracy drops when using fabricated masks, compared to the fabricated Gabor mask baseline, as shown in the comparison (over 10 runs) below. Furthermore, we would like to clarify that the large accuracy drop of 4% could be attributed to non-idealities in in-house fabrication and other experimental errors such as mask-sensor alignments. Such accuracy drops between simulation and in-house fabricated systems have been observed before. For example, in “All-optical machine learning using diffractive deep neural networks” (X.Lin, Science’18) only 88% of the correctly classified images by the optimal model can be still correctly classified after real fabrication on the small scale dataset MNIST. It can be expected that with industry-standard fabrication and manufacturing quality, the resulting accuracy drops can be alleviated.
>
> |  | Accuracy (%) (variance)    |
> |:-:|:-:|
> | Gabor | 87.17 (0.34) |
> | **SACoD** | **89.84 (0.03)** |
>
> **2. Empirical comparison about the designs in Figure 3.**
>
> Yes, we have empirically evaluated the three designs in Appendix C to justify the reason why we pick Figure 3(a) as our mask design for all the experiments. In particular, we find that the design of Figure 3(a) achieves both improved accuracy and efficiency among the three design choices based on the results introduced in Appendix C. We will further clarify this in our final version.
>
> **3. Evaluation of SACoD on IoT systems.**
>
> To verify if the efficiency improvement of SACoD can meet the need of real-world IoT systems, we measure the energy cost of the models in Figure 4 of our manuscript on a Raspberry Pi 4 (Raspi 4) using the following pipeline: (1) convert the models into a TensorFlow Lite (TFLite) format, and (2) optimize the implementation using the official interpreter (Google LLC., 2020) in Raspi 4. From the table below we can observe that SACoD can consistently achieve a better accuracy and energy trade-off, e.g., a 5.46% higher accuracy with 2.29x less energy consumption on CIFAR-100. Note that the 156.4mJ inference energy consumption per image of SACoD (with an accuracy of 74.32%) can satisfy requirements of general IoT systems, for which we provide a reference “the 339mJ inference energy of VGG-16 when executed on a SOTA dedicated DNN accelerator chip” published as “Eyeriss: An Energy-Efficient Reconfigurable Accelerator for Deep Convolutional Neural Networks” (Y. Chen, JSSC’17). Note that SACoD systems’ energy consumption can be much reduced to fit the energy constraints of more stringent IoT devices/applications, if SOTA DNN acceleration designs are leveraged to implement SACoD generated backend models, e.g., using analog and mixed signal integrated circuits developed in “An Always-On 3.8 μJ/86% CIFAR-10 Mixed-Signal Binary CNN Processor With All Memory on Chip in 28-nm CMOS” (D. Bankman, JSSC’19).
>
> ***SACoD on CIFAR-100:***
>
> | Index (diff. trade-offs)| 1     | 2     | 3     | 4     | 5     |
> |:-:|:-:|:-:|:-:|:-:|:-:|
> | Acc (%)         | 76.60 | 76.24 | 75.29 | 74.50 | 74.32 |
> | Energy on Raspi 4 (mJ) |334.3|213.3|182.3| 159.5| 156.4 |
>
> ***Gabor on CIFAR-100:***
>
> | Index (diff. trade-offs)| 1     | 2     | 3     | 4     | 5     |
> |:-:|:-:|:-:|:-:|:-:|:-:|
> | Acc (%)         | 68.86 | 68.33 | 67.54 | 66.37 | 66.31 |
> | Energy on Raspi 4 (mJ) | 357.4 | 214.8 |177.5 | 153.5| 151.7 |

---

> ### Author Response · Authors · 2020-11-25
> **Response to Reviewer 1: Part 2**
>
> (Here's the Part 2 of our responses, see Part 1 for more responses.)
>
> **4. Comparisons with regular CNNs and other types of masks.**
>
> Thanks for the suggestion. First, we benchmark with regular CNN without optical layers, i.e., the lens-based systems, in Appendix D. We find that under a slightly smaller FLOPs (154M FLOPs vs. 158M FLOPs), SACoD achieves a slightly lower accuracy (0.39% and 0.62%) on CIFAR-10 and CIFAR-100, respectively, while reducing the thickness of the imaging systems by 10×, making it applicable to IoT applications with stringent thickness constraints. In general, our SACoD generated models in all our experiments can achieve a comparable or slightly degraded accuracy-complexity trade-off (accuracy drops <0.8% under the same/comparable FLOPs/Energy) as that of their lens-based counterparts, while offering over one order (lens vs. PhlatCam) of thickness reduction in the front-end sensor.
>
> Second, we mainly consider the Gabor mask since they are general-purpose and evident from the fact that they appear in the first layer of most image-based neural networks (“How transferable are features in deep neural networks?” (J. Yosinski, NeurIPS’04) and “ASP vision: Optically computing the first layer of convolutional neural networks using angle sensitive pixels.”  (H. Chen, CVPR’16)).  Currently, we are unaware of other mask filters with such a strong backing. We would be happy to include baselines with other masks in our final version, if you have recommended ones.
>
> **5. Less increase on CIFAR-10 than CIFAR-100 in Table 2.**
>
> As widely observed by previous works (e.g., Table 3 in “Training Very Deep Networks” (R. Srivastava, NeurIPS’15)), CIFAR-100 contains more classes with more complex input statistics than CIFAR-10, which can make it more sensitive to the changes of  different techniques, as verified by Figure 4 of our manuscript.
>
> **6. Too small font size in Figure 4 and 6.**
>
> Thanks for pointing out and we will revise in the final version.

---

### Official Review · AnonReviewer2 · 2020-10-27
**Review for Submission 2602**

**Rating:** 6
**Confidence:** 1

**Review:**

##########################################################################
Summary:

SACoD presents a novel attempt to integrate the computational capabilities of a lensless imaging system, PhlatCam, with the search for the optimal convolutional neural network design for a given task. SACoD provides a framework which enables joint optimization of sensor and CNN resulting in IoT devices that achieve higher task accuracy’s with limited resource budgets of a typical IoT system. The authors present a new an optical layer design that enables above described features. Detailed experiments comparing SACoD sensor + CNN with other baseline models covering past papers, demonstrate the superiority of SACoD’s accuracy/efficiency curve over that of separately optimizing CNN arch or sensor/CNN joint-optimizations that do not vary network architecture. Additionally, ablation studies and results from measurements from actual phase masks fabricated help breakdown the accuracy/efficiency benefits of SACoD while analyzing the noise limitations of mask fabrication process.

##########################################################################
Reasons for score:

The paper presents a sound theoretical description of the SACoD approach along with their optical layer design. The results presented through experiments clearly show the superiority of SACoD approach over other baselines that do not utilize the cost-free computational capabilities of the PhlatCam sensor. With the concerns raised in the Cons, sections answered I recommend that this paper be accepted.

##########################################################################
Pros:

1.	Section 2 presents a good overview of the different approaches attempting sensor/CNN optimization to improve accuracy for given hardware resources. The proposed solution is unique in attempting to utilize the computation capability of PhlatCam imaging system.
2.	Section 3 presents a good theoretical description of the SACoD framework including the steps involved in its training to estimate the optimal mask, network arch and weights.
3.	It is commendable that as noted at the end of Section 3, authors try out both differentiable and reinforcement learning (RL) based NAS approaches and achieve similar accuracies. Highlighting, the effectiveness of SACoD irrespective of the NAS approach utilized.
4.	Section 4 describes the detailed experiments that the authors used to assess the accuracy/efficiency benefits of SACoD over two other baseline approaches for 6 datasets and 4 tasks. Figures 4/5/6 are clear and help demonstrate the superior accuracy vs efficiency curves plotted for SACoD.
5.	Ablation studies discussed in Section 4.6 on mask flexibility influence and optical layer effectiveness are useful to illustrate the importance of joint optimization of sensor and network as well as the effectiveness of SACoD optical layer.
6.	Figures showing a comparison of the performance of different approaches on vision tasks are helpful to show the superiority of SACoD.

##########################################################################
Cons:

1.	Section 3, The optical sensing frontend subsection, describes how the object that is being imaged is at a distance d to the camera while trying to formulate the output of the masks in terms of a 2D convolution operation. Further in this subsection, the authors note that the mask is fixed at a distance d from the sensor. It would be useful to rephrase either of these sentences to clarify what is the distances of the object from the mask and the sensor.
2.	Authors make an effort to describe the impact α on w* and m*, it is not obvious to me as to why the dependence changes for w* and m*. Perhaps an example illustrating the indirect influence of α on m* would be help clarify this.
3.	The experiments in Section 4, present a range of datapoints for SACoD and other baselines for each task/benchmark. In my understanding, these points are obtained by tuning the accuracy vs efficiency tradeoff. How does the size of network change with these experiments? I worry that the efficiency of the GPU utilized for Flops and energy measurements might have biased the results somewhat against SACoD. Testing on a hardware platform with smaller on-chip memory and parallelism might be interesting exercise to show SACoDs superiority.
4.	Section 4.5 is commendable for confirming through measurements that the desired PSF can indeed be generated through fabricated masks. Further the ablation studies discussed in section 4.6 and appendix attempt to model the noise added due to fabrication process in the masks and the resultant accuracy loss.
5.	Its not clear that the magnitude of normal noise assumed by the authors is sufficiently capturing the fabrication noise measured. Considering that the accuracy drops reported in Table 1 and Figure 10 are for different benchmarks. It would be great if authors could use the same benchmark to model the measured 4% reduction in accuracy and then compare with other baselines. Cause for CIFAR 10, the SACoD approach does not have enough margin to dominate Gabor mask baseline despite a 4% accuracy loss.
6.	This dependence on the noise during fabrication is a critical weakness of the SACoD approach. Since it relies on the computations carried out by these potentially noisy masks. The authors should consider developing a full-proof solution to this problem by utilizing noise model based training for their models such that the final network can be immune to the fabrication noises.

##########################################################################
Questions during rebuttal period:

Kindly address the concerns noted in the Cons section

---

> ### Author Response · Authors · 2020-11-25
> **Response to Reviewer 2**
>
> Thank you very much for recognizing our work with “a sound theoretical description of the SACoD approach and experiments clearly show the superiority of SACoD” and for your insightful suggestions. Below are our answers to you questions/comments:
>
> **1. The distance of the object from the mask and the sensor.**
>
> Sorry for the confusion here. The point spread-functions (PSF) is determined by the distance between the mask and sensor, annotated as d, which is the key form-factor/thickness for lensless systems. Once the PSF is optimized and the mask is fabricated, the mask will be fixed at the chosen d, which is slightly less than 2mm from the CMOS sensor in our system. And the distance between the object and mask in our experimental systems is about 11 inches. We will clarify the two distances in the final version.
>
> **2. The influence of $\alpha$ on $m^\*$.**
>
> Thanks for the suggestion! $\alpha$ controls the searched network structure, which favors different distributions of phase masks $m^*$. In our physical measurement experiments, we do observe that the optimal point spread-functions (PSF) of the phase masks for different searched networks are quite different and we will utilize such examples to illustrate the influence of $\alpha$ on $m^*$ in the final version.
>
> **3. Model size changes in Sec. 4 and measurement on platforms with smaller on-chip memories.**
>
> Thanks for the suggestion. Yes, you are right about “these points are obtained by tuning the accuracy vs efficiency tradeoff”, i.e., $\lambda$ in Eq. (3) of our manuscript. Specifically,  a larger $\lambda$ enforces the SACoD search to prioritize efficiency, thus resulting in a smaller network size with a lower accuracy; while a smaller $\lambda$ guides SACoD’s search to favor a larger network size with a higher accuracy. Meanwhile, we can observe from Fig. 4 of our manuscript and the experiments below that there exists a high correlation between the networks’ FLOPs and model sizes.
>
> Following your suggestion, we conduct extra experiments to  measure the inference energy cost per image on Raspberry Pi 4 (Raspi 4) using the following pipeline: (1) convert the models into a TensorFlow Lite (TFLite) format, and (2) optimize the implementation using the official interpreter (Google LLC., 2020) in Raspi 4. From five runs of experiment results (each targets a different accuracy efficiency trade-off) in the table below, we can observe that SACoD again consistently achieves a better accuracy and energy trade-off, e.g., a 5.46% higher accuracy with 2.29x less energy consumption, verifying that SACoD is generally applicable on different devices.
>
> ***SACoD on CIFAR-100:***
>
> | Index (diff. trade-offs) | 1  | 2     | 3     | 4     | 5     |
> |:-:|:-:|:-:|:-:|:-:|:-:|
> | Acc (%)         | 76.60 | 76.24 | 75.29 | 74.50 | 74.32 |
> | MFLOPs          | 158.9 |  95.2 |  80.6 |  66.4 |  57.5 |
> | Model size (MB) | 3.70  | 2.18  | 2.39  | 1.93  | 1.82  |
> | Energy on Raspi 4 (mJ) |334.3|213.3|182.3| 159.5| 156.4 |
>
> ***Gabor on CIFAR-100:***
>
> | Index (diff. trade-offs) | 1     | 2     | 3     | 4     | 5     |
> |:-:|:-:|:-:|:-:|:-:|:-:|
> | Acc (%)         | 68.86 | 68.33 | 67.54 | 66.37 | 66.31 |
> | MFLOPs         | 160.5  |  97.8 | 80.9  |  65.3 |  58.2 |
> | Model size (MB) | 3.60  | 2.21  | 2.15  | 1.97  | 1.85  |
> | Energy on Raspi 4 (mJ) | 357.4 | 214.8 |177.5 | 153.5| 151.7 |
>
> **4. Comparison with baselines under noise using the same benchmark.**
>
> Thanks for the suggestion. Following your suggestion to fairly benchmark with the baselines, we fabricated the Gabor mask and tested its achieved accuracy following the same setting as that for  Table 1 of our manuscript. We can observe that SACoD generated designs lead to a lower accuracy drop when using fabricated masks,  compared to the Gabor mask baseline, as shown in the comparison (over 10 runs) in the table below.
>
> |  | Accuracy (%)    |
> |:-:|:-:|
> | Gabor | 87.17 (0.34) |
> | **SACoD** | **89.84 (0.03)** |
>
> Furthermore, we would like to clarify that the large accuracy drop of 4% could be attributed to non-idealities in in-house fabrication and other experimental errors such as mask-sensor alignments. Such accuracy drops between simulation and in-house fabricated systems have been observed before. For example, in “All-optical machine learning using diffractive deep neural networks” (X.Lin, Science’18) only 88% of the correctly classified images by the optimal model can be still correctly classified after real fabrication on the small scale dataset MNIST. We can expect that with industry-standard fabrication and manufacturing quality, the resulting accuracy drops can be alleviated.
>
>
> **5.Noise model based training.**
>
> Thank you for your great suggestions of “utilizing noise model based training”. We will build a more comprehensive noise model from the distribution of real fabricated masks, and then integrate it into our method in our future work, as we don’t have sufficient time to conduct the experiments this time.

---

### Official Review · AnonReviewer4 · 2020-10-30
**An interesting idea with some issues**

**Rating:** 6
**Confidence:** 4

**Review:**

#################################

Summary:

The paper proposed to adopt differentiable network architecture search (DARTS) for the co-design of the sensor (a lensless camera) and the deep model for visual recognition tasks, so as to maximize the accuracy and minimize the energy consumption. The key idea is to include the sensor configuration, in this case the phase mask of a lensless camera modeled as 2D convolutions, as additional parameters in architecture search.  The proposed method was evaluated on simulated data for a number of vision tasks (image classification, face recognition and head pose estimation), as well as using fabricated masks on a real world camera. The results demonstrated significantly increase recognition performance given the same energy level.

#################################

Pros:
* The high-level idea of using NAS for the co-design of the sensor and the deep model is quite exciting.
* The experiments are extensive, including both several vision tasks in simulation and a classification task using real-world sensor implementation.

#################################

Cons:
* Sensor Design: Generality vs Specificity

The paper proposed to tailor a physical sensor and bundle it with a deep model for a target vision task. Once realized, it is unclear if the sensor is able to adapt to a different task or even a different model. Would the phase mask identified on one model / task generalize to different tasks/models? Generality is an important property of visual sensors (e.g., RGB / ToF cameras). The same camera can be used for different tasks (e.g., image classification, face recognition, etc). And the captured images can be examined by various models. It is probably not surprising that higher accuracy and lower energy can be achieved if a sensor is specifically designed for a single task and for a single type of model. The question is why do we need such a high specificity. It seems very limited even in the IoT setting. For example, what if the backend model needs to be updated.

* Lack of modeling for fabrication error

Fabrication error is quite significant and is not modeled. For example, on CIFAR 10, fabrication error accounts for a over 4% drop in accuracy, canceling out most of the gains from the co-design (using a Gabor mask has 2-3% drop in accuracy in comparison to the co-design). Is it possible to model fabrication error and encode that into the loss function for NAS? For example, a good design should avoid those patterns that are likely to create issues in fabrication.

#################################

Minor comments:

The paper has a strong vision flavor and might be a better fit to vision / computational photography conferences.

#################################

Justification for score:

Overall, this is a paper with a very interesting idea and solid experiments. Yet, the problem setting is limited and the modeling has some issues. I am more positive about this paper after reading the authors' response.

---

> ### Author Response · Authors · 2020-11-25
> **Response to Reviewer 4**
>
> Thank you very much for recognizing our work with “this is a paper with a very interesting idea and solid experiments” and for your insightful questions/comments. Below we provide our detailed response:
>
> **1. Generality vs Specificity.**
>
> Good questions! First, we answer the question of generalization to backend models. We conduct additional experiments to show that SACoD’s generated masks feature better generalization capability, as summarized in the following table. In particular, we transfer the SACoD generated mask (optimized on datasets of FlatCam Face) to MobileNetV2, and compare the achieved accuracy on the same network with a gabor-mask. It is shown that the SACoD generated mask achieves a 1.6% and 4.3% higher accuracy than the gabor-mask on CIFAR-10 and CIFAR-100, respectively, validating its better generalization capability.
>
> |Mask Design|Accuracy(%) on CIFAR-10 | Accuracy(%) on CIFAR-100|
> |:-:|:-:|:-:|
> |Gabor|91.5| 68.2|
> |**SACoD optimized on FlatCam Face**|**93.1**|**72.5**|
>
> Second, under resource constraints such as FLOPs, increased task-specificity allows achieving performances (similar to a generic system, e.g., comparisons with the lens-based system in Appendix D). We further empirically validate the task-specificity: we manufacture the mask optimized on CIFAR-100 and then transfer it to the Flatcam Face task as introduced in Sec. 4.1, to compare the resulting accuracy with that of the manufactured mask directly customized/specialized for the Flatcam Face task. We repeat the manufacture process for five times and report the average accuracy and variance in the table below. We can see that the customized mask can achieve a 1.81% higher averaged accuracy, while having a slightly reduced/comparable variance, which is a nontrivial improvement for face-related tasks in real-world applications.  This set of experiments validate the advantage of specificity. Furthermore, a face-related mask could be generalized to other face-related applications. By this, we can maintain generalization for face-based applications while achieving high performance by being face-specific. We will explore this in our future work.
>
> |Mask Design|Accuracy (%) on Flatcam Face  (variance)|
> |:-:|:-:|
> |Mask optimized on CIFAR-100|95.31 (0.017)|
> |**Mask optimized on Flatcam Face**| **97.12 (0.015)**|
>
> In addition, we agree that higher accuracy and lower energy can be achieved if a sensor is specifically designed for a single task and for a single type of model, which is often the motivation for high specificity yet might not be affordable especially for IoT applications, due to its corresponding high costs. One key highlight of our SACoD is that it achieves such high specificity at extremely low costs, as each mask costs one order of magnitude lower than lens-based cameras (see Figure 2 of “FlatCam: Thin, Bare-Sensor Cameras using Coded Aperture and Computation” (M. Asif, IEEE Trans. Comput. Imag.’16)), in addition to FlatCam’s advantageous thin feature.
>
> As such, we can see that our SACoD demonstrates not only the first sensor-network co-search framework, but also its generated designs have the benefits of high speciality without having the associated high costs. Furthermore, SACoD’s generated masks have better generalization capability. From these perspectives, SACoD pushes forward the pareto curve of specificity-cost trade-offs.
>
> **2. Lack of modeling for fabrication error.**
>
> Thank you for your great suggestions of “model fabrication error and encode that into the loss function for NAS”!  First, we would like to update you that SACoD generated designs lead to LOWER accuracy drops when using fabricated optimized-masks, compared to the fabricated gabor-mask baseline, as shown in the comparison (over 10 runs) below based on two datasets and models. We can see that while both SACoD generated designs and the gabor-mask baselines suffer from accuracy losses due to the non-idealities of fabricated masks, our SACoD’s generated designs consistently lead to lower accuracy drops.
>
> |Mask Design|Accuracy (%) (variance)|
> |:-:|:-:|
> |Gabor|87.17 (0.34)|
> |**SACoD**|**89.84 (0.03)**|
>
> Second, we would like to clarify that the large accuracy drop of 4% could be attributed to non-idealities in in-house fabrication and other experimental errors such as mask-sensor alignments. Such accuracy drops between simulation and in-house fabricated systems have been observed before. For example, in “All-optical machine learning using diffractive deep neural networks.” (X.Lin, Science’18) only 88% of the correctly classified images by the optimal model can be still correctly classified after real fabrication on the small scale dataset MNIST. It can be expected that with industry-standard fabrication and manufacturing quality, the resulting accuracy drops can be alleviated. Finally, your suggested idea is very interesting, which we will explore as our next future work, as we don’t have sufficient time to conduct the experiments this time.

---

### Official Review · AnonReviewer3 · 2020-10-31
**This paper presents a joint optimization on signal acquisition front-end and back-end CNN algorithms to achieve deep learning's practical significance on edge devices.**

**Rating:** 6
**Confidence:** 4

**Review:**

The paper addresses the practical application-level problem with joint optimization from front-end sensor to back-end CNN algorithms. Authors validate the proposed method's advantages with comprehensive experiments on different tasks, including image classification, face/pose detection, image segmentation, and image translation. Lastly, the authors claim to contribute its source codes to the research community upon acceptance. Overall, I think this research paper is well-written with no flaw identified; more importantly, I believe this investigation will substantially impact many real-world applications via pushing CNN to edge devices.
One suggestion is to specify the search space for NAS, as the search space is usually application dependent. As it is not specified in the paper, I assume you adopt the default search space, which contains only the 10 operations, and I wonder what is the best set of operations among them that best for the sensor data?

---

> ### Author Response · Authors · 2020-11-25
> **Response to Reviewer 3**
>
> Thank you for recognizing the importance of our target problem and for your valuable suggestions. Below are our answers to you questions:
>
> **1. Detailed search spaces and the best set of operations for the sensor data.**
>
> We discuss the details of the adopted search spaces for the classification, image translation, and segmentation tasks in our Appendix A, E1, and E4, respectively. We will add more details in the main text of the final version.
>
> Here we take the classification tasks as an example to further elaborate: we adopt the search space in FBNet (B. Wu, CVPR’19) with nine operations: [*k3e1g2*, *k3e1*, *k3e3*, *k3e6*, *k5e1g2*, *k5e1*, *k5e3*, *k5e6*, *skip*] where *k* is the kernel size, *e* is the expansion, *g* is the group, and *skip* denotes the skip connection. As such, *k5e1g2* denotes a convolution with a kernel size of 5, a channel expansion of 1, and a group of 2. We find that the best set of searched operations are *k3e6*, *k5e6*, and *skip* connections, showing that the searched networks tend to be wide and shallow networks. This indicates that for the sensor data, the searched networks favor larger channel numbers for better extracting the features, while leveraging skip connections to trade for efficiency.

---

### Decision · Program_Chairs · 2021-01-07
**Final Decision**

**Decision:**

Reject

**Comment:**

This paper attempts to jointly search for the sensor and the neural network architecture. More specifically, the proposed approach jointly optimizes the parameters governing the PhlatCam sensor and the backend CNN model. In terms of the approach, the paper follows a well known DARTS formulation for the differentiable architecture search. A very straightforward solution was proposed for the problem.

Although all the reviewers place that the paper is marginally above the acceptance threshold, none of them strongly support the paper and the reviewers point out that the paper is limited in terms of the setting and data. The problem formulation of the paper itself is interesting, but the AC agrees with the reviewers that the paper is limited and lacks enough technical contributions to warrant the acceptance to ICLR.